# A Nuclear Magnetic Resonance (NMR)- and Mass Spectrometry (MS)-Based Saturation Kinetics Model of a *Bryophyllum pinnatum* Decoction as a Treatment for Kidney Stones

**DOI:** 10.3390/ijms25105280

**Published:** 2024-05-12

**Authors:** Candus Chik, Anne-Laure Larroque, Yuan Zhuang, Shane Feinstein, Donald L. Smith, Sero Andonian, Aimee K. Ryan, Bertrand Jean-Claude, Indra R. Gupta

**Affiliations:** 1Department of Human Genetics, McGill University, Montreal, QC H3A 0C7, Canada; 2The Research Institute of the McGill University Health Center, Montreal, QC H4A 3J1, Canada; 3Plant Science Department, McDonald Campus, McGill University, Sainte-Anne-de-Bellevue, QC H9X 3V9, Canada; 4Division of Urology, McGill University, Montreal, QC H4A 3J1, Canada; 5Department of Pediatrics, McGill University, Montreal, QC H4A 3J1, Canada; 6Department of Medicine, Division of Medical Oncology, McGill University, Montreal, QC H4A 3J1, Canada

**Keywords:** kidney stone, nephrolithiasis, citrate, malate, gallate, flavonoids, quercetin

## Abstract

*Bryophyllum pinnatum* (BP) is a medicinal plant used to treat many conditions when taken as a leaf juice, leaves in capsules, as an ethanolic extract, and as herbal tea. These preparations have been chemically analyzed except for decoctions derived from boiled green leaves. In preparation for a clinical trial to validate BP tea as a treatment for kidney stones, we used NMR and MS analyses to characterize the saturation kinetics of the release of metabolites. During boiling of the leaves, (a) the pH decreased to 4.8 within 14 min and then stabilized; (b) regarding organic acids, citric and malic acid were released with maximum release time (t_max_) = 35 min; (c) for glycoflavonoids, quercetin 3-O-α-L-arabinopyranosyl-(1 → 2)-α-L-rhamnopyranoside (Q-3O-ArRh), myricetin 3-O-α-L-arabinopyranosyl-(1 → 2)-α-L-rhamnopyranoside (M-3O-ArRh), kappinatoside, myricitrin, and quercitrin were released with t_max_ = 5–10 min; and (d) the total phenolic content (TPC) and the total antioxidant capacity (TAC) reached a t_max_ at 55 min and 61 min, respectively. In summary, 24 g of leaves boiled in 250 mL of water for 61 min ensures a maximal release of key water-soluble metabolites, including organic acids and flavonoids. These metabolites are beneficial for treating kidney stones because they target oxidative stress and inflammation and inhibit stone formation.

## 1. Introduction

Plant-based natural products have been used for their medicinal properties for decades. Their molecular characterization has inspired drug design and discovery through structure–activity relationship studies for targets identified in many diseases [1]. According to the standards of regulatory agencies, the active molecules have been isolated and evaluated as single agents, based upon their pharmacokinetics, maximum tolerated doses, and pharmacodynamics. The contribution of these single agents to the clinical management of complex diseases is now inestimable. In parallel with the rapid pace of modern drug discovery and development, herbal medicines continue to be taken by communities around the world to manage chronic and non-chronic illnesses. While, in some cases, the active molecules have been identified, most have not been well-characterized nor validated in clinical trials. Herbal medicine does not rely on isolated single substances but is based on formulations that preserve multiple active molecules. Faithful reproduction of herbal medicine formulations is thus a challenge when evaluating their efficacy in clinical trials. The formulations are concoctions, decoctions, plant juices, ethanolic extractions, aqueous extracts, and teas that represent combinations of organic molecules, including carboxylic acids, saccharides, flavonoids, terpenes, saponins, etc. Thus, in contrast to conventional drugs, they contain multiple known and unknown ingredients/dose. Therefore, the clinical testing of these formulations is de facto a combination therapy. Due to their complexity, the entire non-fractionated mixture must be evaluated and standardized according to its molecular signature and physical properties.

Amongst all formulations, concoctions from boiled leaves are the easiest preparation to reproduce. Paradoxically, over the past two decades, there have been very few examples of clinical trials reported with aqueous concoctions. Green tea is one of the most investigated herbal medicines [2,3]. A recent randomized clinical trial with green tea demonstrated its ability to prevent diabetes and its complications [4]. The latter study used tea bags that were boiled for 6 min and administered in volumes defined as cup units [4]. While the study defined key flavones in the mixture, there was no rationale for the 6 min boiling time, the pH of the tea was not defined, and there was a lack of precision in the tea volumes recommended per participant.

Here, we evaluate *Bryophyllum pinnatum* (BP), a perennial herb also known as *Kalanchoe pinnata* and by many common names, including Air Plant, Miracle Leaf, Leaf of Life, and Cathedral Bells [5,6]. It is a drought-tolerant, succulent member of the Crassulaceae family that thrives in rocky soil and warm climates [5,6]. BP, like all species in the genus *Bryophyllum*, is originally from Madagascar but has spread throughout the world as a houseplant and ornamental garden plant.

The leaves of BP are prescribed in decoction or poultice form by practitioners of herbal medicine and are used to treat a variety of ailments, including infections, kidney and urinary disorders, dermatological conditions, snake bites, hypertension, sleep disturbances, and premature labor. The leaves are given as a leaf juice, leaves in pills, and as alcoholic extracts [7,8,9,10]. These formulations of BP have also been demonstrated in laboratory and pilot clinical trials to have antiurolithiatic activity [11,12,13,14,15]. In a rat kidney stone model, aqueous, alcoholic, and non-alcoholic extracts of BP leaves prevented the formation of kidney stones and reduced the size of existing stones [11,12,13]. In an uncontrolled trial, patients with renal stones who were treated with BP juice either passed their stones or showed a reduction in the size of their stones [15]. Current treatment options for individuals with recurrent kidney stones are extremely limited and require long-term adherence. Patients are asked to follow the ‘stone prevention diet’, which consists of high water intake and low dietary sodium and protein [16]. However, many have difficulty maintaining this diet [16]. Thiazide diuretics are widely used to treat and prevent recurrent kidney stones, but they have significant side effects and may not be effective [17]. There is therefore great interest in exploring new alternative treatments, including the use of BP.

Despite the fact that BP decoction is an herbal medicine used to treat a variety of ailments, this formulation has not undergone clinical evaluation [15]. Here, we define a decoction formulation of BP using ^1^H NMR and LC-MS/MS that is based upon the kinetics of the release of key metabolites during boiling. The decoction yields the optimal metabolite profile for use as a treatment for kidney stones.

## 2. Results

### 2.1. pH Measurement

The pH of the decoction decreased as a function of time and followed first-order kinetics at 100 °C (Figure 1). The pH dropped rapidly and stabilized at 4.8 within 14 min of boiling. The rapid increase in acidity of the decoction suggested the release of multiple organic acids from the boiled leaves.

### 2.2. Identification of Metabolites

^1^H NMR and MS analyses of the decoction confirmed the presence of the following organic acids: citric acid, malic acid, and gallic acid (see structures, Figure A1, Figure 2 and Figure 3). To confirm the identity of citric acid and malic acid, spiking with their respective commercially available samples was performed [18]. Changes in peak intensity confirmed their identity based on chemical shift assignment (Figure 2). Malic acid, a dicarboxylic acid, and citric acid, a tricarboxylic acid, appeared as overlapping peaks between 2.38 and 2.85 ppm. Their kinetics, when measured together, revealed a t_max_ = 35 min (Figure 3). The release of gallic acid, a common phenolic acid observed in plants, followed a slow kinetics of release (Figure 4). The predicted t_max_ values of the saturation curves, determined by ^1^H NMR and further confirmed by LC-MS/MS, were ~13 h and ~10 h, respectively. Gallic acid was identified in the aromatic region by ^1^H NMR as a singlet at 7.05 ppm and in LC-MS/MS at *m*/*z* 169 [M−H]^−^.

The analysis of flavone conjugates was performed using MS-MS fingerprints for structure assignment and revealed the following flavonoids: quercetin 3-O-α-L-arabinopyranosyl-(1 → 2)-α-L-rhamnopyranoside (Q-3O-ArRh), myricetin 3-O-α-L-arabinopyranosyl-(1 → 2)-α-L-rhamnopyranoside (M-3O-ArRh), kaempferol 3-O-α-L-arabinopyranosyl (1 → 2)-α-L-rhamnopyranoside (kappinatoside), myricitrin, and quercitrin (see structures, Figure A1) at *m*/*z* 579 [M−H]^−^, 595 [M−H]^−^, 563 [M−H]^−^, 463 [M−H]^−^, and 447 [M−H]^−^, respectively. The disaccharide-conjugated flavonoids, Q-3O-ArRh, M-3O-ArRh, and kappinatoside, were rapidly released into the decoction according to kinetics that revealed saturation after approximately 10, 7, and 5 min of boiling, respectively (Figure 5). Two monosaccharide-conjugated flavonoids, myricitrin and quercitrin, were rapidly released within the first 5 min followed by a decrease over the period of analysis (Figure 6).

### 2.3. Total Phenolic Content and Total Antioxidant Capacity

Phenolic compounds are major components of plants and exhibit free radical scavenging activity to reduce oxidative stress [19]. Our results indicate that their kinetics of release vary from fast (flavonoids) to slow rates (gallic acid). While we measured the kinetics of selected phenolic compounds, the composition of decoctions includes a wide variety of phenolic metabolites that contribute to their total antioxidant activities. Thus, we determined the optimal boiling time for the maximal release of the total phenolic content of the decoction. The total phenolic content (TPC) of the decoction was measured as catechin equivalent using an azo-coupling-based colorimetric assay. The TPC in the decoction exhibited a saturation curve with a t_max_ of 55 min at a concentration of 0.4 mM catechin equivalent (Figure 7a).

The direct and well-established correlation between TPC and the total antioxidant capacity (TAC) of natural products provided a rationale for determining the t_max_ of the latter parameter in the context of this study. TAC was measured using a copper-reduction-based colorimetric assay. The release of TAC followed a saturation kinetics profile, reaching a t_max_ of 61 min at 4.7 mM Trolox equivalent (Figure 7b).

### 2.4. The Metabolic Signature of the Decoction

After boiling 24 g of BP leaves in 250 mL of water for a duration of 4 h, we were able to define a metabolic signature of the decoction that can be used as quality control in studies designed to evaluate its antiurolithiatic activity (Figure 8, Figure A1 and Figure A2). Malic acid, citric acid, and gallic acid were present and are known to inhibit stone formation [20,21,22]. Flavonoids, including myricitrin, quercitrin, Q-3O-ArRh, kappinatoside, and M-3O-ArRh, that inhibit oxidative stress and inflammation were also identified. The t_max_ of TPC and TAC was shorter than that of gallic acid, suggesting that gallic acid is not a dominant compound in the total phenolic content (Figure A2). Therefore, considering the TPC and TAC of the decoction, we conclude that a boiling time of 61 min at 100 °C will generate a formulation with an optimal metabolic signature that includes key antiurolithiatic metabolites (Figure 8).

## 3. Discussion

Boiling leaves is a decoction that drives water-soluble metabolites into the aqueous phase according to rates defined by their solubility, stability, temperature, and pH. To evaluate a decoction made from *Bryophyllum pinnatum* (BP) leaves as a treatment for kidney stones, we boiled 24 g of leaves in 250 mL of water and used ^1^H NMR and LC-MS/MS to analyze its contents. We established that the optimal metabolic signature was present after 61 min of boiling. The decoction was mildly acidic due to the presence of multiple carboxylic acids that were still detected after 4 h of boiling. Likewise, the flavone metabolites reached a plateau before 10 min that was sustained up to 4 h. The carboxylic acids and the flavonoids have known medicinal properties that are beneficial for treating kidney stones [23,24].

NMR and MS analyses have previously been used on formulations of BP such as alcoholic extracts, aqueous extracts, and leaf juice to identify the plant’s medicinal components [25,26,27]. Consistent with our results, these studies have shown that the plant is mainly composed of organic acids, flavonoids, and phenolic compounds [25,26,27]. As expected, many non-polar compounds found in alcoholic extracts were not present in aqueous extracts [25,26,27]. When comparing cold versus hot aqueous extractions from the leaves, the hot aqueous extracts are more favorable for flavonoid extraction, while the cold extracts generate a more diverse set of organic acids [28]. While these studies achieved authentication and identification of markers of BP, they have not been used to standardize preparations nor to analyze metabolites at boiling temperatures in a decoction of BP.

We observed that the phenol and flavonoid signature was stable over prolonged boiling at 100 °C for up to 240 min. This is consistent with several studies that confirmed the presence or stability of phenolic compounds at high temperatures (e.g., catechins and gallic acid) [29,30]. A recent study reporting on the degradation of free quercetin when dissolved in water at 100 °C for up to 180 min demonstrated that it is hydrolyzed into other phenolic compounds, such as 3,4-dihydroxyphenylglyoxylate, 2,4,6-trihydroxybenzoic acid, 3,4-dihydroxybenzoic acid, and 3,4,5-trihydroxybenzoic acid (gallic acid) [31]. With the exception of gallic acid, none of the reported metabolites were detected in our BP decoction.

Previous studies have reported the presence of bufadienolides in dried and methanolic extracts of BP leaves, specifically bersaldegenin-1-acetate, bersaldegenin-3-acetate, bryophyllin A, and bersaldegenin-1,3,5-orthoacetate [32,33]. These are known to be cytotoxic at higher concentrations [32,33]. We did not detect MS/MS peaks corresponding to these metabolites. This may be due to the fact that they are unstable under boiling conditions or that they are present at undetectable concentrations. In contrast to methanolic extracts that capture both water-soluble and liposoluble metabolites, decoctions are solutions of water-soluble molecules, which are easier to eliminate in the urine and/or in bile, thereby reducing the risk of systemic tissue toxicity.

Based upon fragmentation patterns, MS allowed for an unequivocal characterization of flavone conjugates, including the rare 3-O-α-L-arabinopyranosyl-(1 → 2)-α-L-rhamnopyranoside moiety—Q-3O-ArRh, M-3O-ArRh, and kappinatoside [25]. Importantly, we identified Q-3O-ArRh, also referred to as Bp1, which has been reported to be the main flavonoid glycoside of BP [25,34,35,36]. To our knowledge, this unique metabolite has not been reported in other plant decoctions. Q-3O-ArRh exerts inhibitory activity against phosphodiesterase 4B (PDE4B), which is involved in inflammation [35,37]. PDE4B hydrolyzes cyclic AMP (cAMP), which is an important second messenger that promotes the production of anti-inflammatory cytokines. Therefore, PDE4B inhibition results in higher levels of cAMP and an anti-inflammatory effect [37].

In addition to the flavones conjugated with disaccharides, we also identified flavones conjugated with monosaccharides: quercitrin and myricitrin. The glycosylation of flavonoids is of great importance since this step can affect intestinal absorption, binding to plasma proteins, and subsequently their pharmacokinetic effects [38]. The kinetics of the release of quercitrin and myricitrin were marked by a rapid release into the decoction, perhaps because they are easily extracted at boiling temperature. Although the biological effects of glycosylated flavonoids are complex, it appears that both myricitrin and quercitrin can scavenge free radicals generated through oxidative stress and thus decrease inflammation [39,40]. Quercitrin also inhibits anti-cholinergic esterase activity that may be beneficial in the treatment of kidney stones by promoting urinary tract smooth muscle relaxation and passage of a kidney stone [41].

While the kinetics of the release of malic acid and citric acid followed a saturation curve similar to the flavones, gallic acid showed a unique profile, with a t_max_ greater than 60 min [21,24,42,43]. This may be due to its slow formation from the hydrolysis of ester bonds in tannins and other gallates such as epigallocatechin during boiling [22,44]. The fact that the kinetics of gallic acid did not influence the overall rate of release of the TPC suggests that it is less abundant when compared to the other phenolic metabolites in the decoction [see gallic acid and flavonoids (Figure A1 and Figure A2)].

The two categories of molecules identified, organic acids and flavonoids, have important biological activities with respect to urolithiasis [23,24]. Individually, malic, citric, and gallic acid have all been reported to exert antiurolithiatic activities [20,21,22]. The flavonoids and phenolic compounds are known to have significant anti-inflammatory and antioxidant properties that are beneficial in the treatment of kidney stones [45,46,47,48,49,50]. Thus, their complete release in the decoction is essential in the formulation for clinical evaluation. As depicted in Figure 8, the t_max_ of the release of the targeted components of the tea varied from <5 to >60 min and remained stable during 4 h of boiling, as evidenced by the saturation phase of the curves. We therefore propose 61 min of boiling to capture all of the metabolites and maximize the antioxidant activity of the decoction. Further work is required to demonstrate the efficacy of the formulation in animal models and in randomized clinical trials.

## 4. Materials and Methods

### 4.1. Source of B. pinnatum

Plants originated in Miami, Florida and were identified as BP by leaf morphology (Appendix A). Green leaves were collected from plants that were grown indoors at the Faculty of Agricultural and Environmental Sciences, McDonald Campus, McGill University at 20 ± 2 °C under 55% relative humidity. The light intensity was approximately 400 μmol m^−2^ s^−1^, supplied from LED lights (VYPR Series, Fluence, Melbourne, VIC, Australia). Plants were watered regularly with nutrient solution, following recommendations provided by Remo Brands Inc. (Maple Ridge, BC, Canada). Two batches of leaves were obtained at different times and stored in a −20 °C freezer for one to three weeks before use. Each batch was thawed at room temperature for 30 min and then used for boiling experiments.

### 4.2. Kinetics of Metabolite Release in the Decoction

Water (250 mL) was heated to boiling (100 °C) under reflux in an oil bath in a 500 mL 3-neck round bottom flask. Leaves (24 g) were cut into small rectangular fragments (approximately 1 cm by 0.5 cm) and added through a solid funnel. The initial zero time point was set upon complete addition of the leaves. Aliquots (1 mL) were taken through a 3 mL syringe at the following time points: 0, 5, 10, 20, 30, 40, 45, 60, 75, 90, 120, 180, 240 min. Temperature was continuously monitored and maintained at 100 °C. The experiment was stopped after 4 h, and the resulting brown solution was cooled and stored at 4 °C. The 1 mL aliquots were stored at −20 °C until analysis.

### 4.3. Kinetics of pH Change

The pH of each sample was measured at room temperature by introducing a micro-electrode (InLab^®^ Micro-Pro-ISM probe, Mettler Toledo) in the 1.5 mL aliquot using a Mettler Toledo pH-meter (Seven compact pH meter, Mettler Toledo, Greifensee, Switzerland) prior to any other measurements.

### 4.4. Identification of Metabolites

Metabolites were characterized by ^1^H NMR and LC-MS/MS. Before analysis, the samples were thawed to RT and spun down at 15,000 rpm for 10 min at RT. For NMR, 450 μL of supernatant was transferred to a new 1.5 mL tube and 50 μL of D_2_O spiked with 50 mM of TSP was added. The mixture was then transferred to a 5 mm NMR tube. For LC-MS, 100 to 200 μL of supernatant was filtered through a PES filter (Captiva PES Premium Syringe Filter, Agilent, Santa Clara, CA, USA) to remove any particles.

### 4.5. ^1^H NMR and MS Materials

All reagents and solvents were LC-MS grade. OmniSolv^®^ acetonitrile (ACN) was sourced from MilliporeSigma (St. Louis, MO, USA), formic acid from Fisher (Fisher Chemical™, Fisher Scientific, Waltham, MA, USA), and ultra-pure deionized water was obtained from the Millipore Milli-Q water system (Millipore MOA, Billerica, MA, USA). Sodium trimethylsilyl propionate (TSP) used as an NMR standard and deuterated water (D_2_O) were sourced from Sigma-Aldrich/MilliporeSigma (St. Louis, MO, USA). ESI-L Low Concentration Tuning Mix used as an MS tune mix standard solution was sourced from Agilent Technologies (Santa Clara, CA, USA).

### 4.6. ^1^H NMR Analysis

All samples were analyzed on an AVANCE III HD 600 MHz NMR Bruker spectrometer equipped with a cryo-probe CPQCI 1H-31P/13C/15N and a SampleJET^TM^ autosampler set to 5 mm shuttle mode with the tubes refrigerated at 6 °C. All analyses were run under automation by IconNMR. In order to suppress the water signal and avoid any baseline distortion following acquisition, the spectra were acquired using the AU program, Bruker pulse program: noesygppr1d (relaxation delay-90°-t1-90°-tm-90°-acquisition, Bruker Biospin GmbH, Ettlingen, Germany) at 300 K. The optimal 90° pulse was calculated for each sample using the automated routine. Tuning and matching was adjusted and the sample was locked to the solvent optimized for H_2_O/D_2_O samples and shimmed using the automated routine of IconNMR. The relaxation delay (RD) and mixing time (tm) were set at 2 s and 10 ms, respectively. A total of 32k complex data points were accumulated for 32 scans over a spectral width of 16 ppm using a 1.70 s acquisition delay. All spectra were then automatically phased, baseline corrected, and referenced to the internal TSP signal at 0 ppm.

### 4.7. LC-MS/MS Analysis

LC-MS analyses were performed with an ESI-QTOF-MS system (Impact II, Bruker, Billerica, MA, USA) coupled with a liquid chromatography system (1290 Infinity, Agilent) and equipped with a reversed-phase Agilent Zorbax SB-C18 column (2.1 mm × 50 mm, 1.8 µm). Mobile phases were water with 0.1% formic acid (A) and acetonitrile with 0.1% formic acid (B). The following gradient elution method was used: 0 to 95% B from 0 to 45 min, a 95% B plateau for 10 min, and a return to initial conditions with equilibration for 10 min. The column temperature was set at 40 °C, and the flow rate was 0.25 mL/min. The samples were kept in an autosampler at 10 °C. Each sample was injected at a volume of 10 μL. The operating parameters of the mass spectrometer were as follows: negative spray voltage at 3500 V, dry temperature at 200 °C, dry gas flow at 8 L/min, and nebulizer at 1 bar. The data were collected with a mass range from 50 to 2500 *m*/*z* with an acquisition rate of 8 Hz and stepping. The Auto MS/MS mode was used to obtain fragment ions. The instrument was calibrated using a Tune mix standard solution before the runs, and, at the beginning of each run, this solution was injected as an internal calibrant for MS spectrum calibration. The LC-MS-acquired data were processed using DataAnalysis Version 5.2 software (Bruker) and Metaboscape Version 2023b software (Bruker).

### 4.8. Total Phenolic Content

The total phenolic content was measured using the commercial Phenolic Compound Assay Kit (Colorimetric) according to the protocol provided by Sigma-Aldrich (MAK365). The assay is based on the reaction of a diazonium salt with the phenolic ring of the compounds (e.g., flavonoids, catechins, etc.) in the mixture to produce an azo dye that can be detected and quantified by UV spectrophotometry. Serial dilution of the decoction was performed to ensure that the readings were within the standard curve. The boiled decoction samples were diluted by a factor of 2. Samples (50 µL) were added to a 96-well plate and filled to 100 µL with ultrapure water. PC probe (20 µL) and PC assay buffer (80 µL) were added to each reaction well. Catechin was used to generate a standard curve to determine the concentration of the phenolic compounds in the mixture. Plates were incubated at room temperature for 10 min with gentle shaking before measuring absorbance at 480 nm. Measurements were performed in duplicates with control wells without PC probe. Vanillic acid (50 nM) was used as a positive control. Data were expressed as mM of catechin equivalent.

### 4.9. Antioxidant Activity

Antioxidant activity was measured using the commercially available Antioxidant Assay Kit obtained from Sigma-Aldrich (MAK334). Total antioxidant capacity (TAC) was determined by measuring the reduction of Cu^2+^ to Cu^+^ by molecules with antioxidant properties in the decoction. Subsequently, Cu^+^ formed a coloured complex with the dye reagent in the kit that was detected and quantified by UV spectrophotometry. Decoction samples were diluted by a factor of 4 with ultrapure water. Samples with (20 µL) were added to a 96-well plate with Reaction mix from the kit (100 µL). Plates were incubated at room temperature for 10 min before measuring absorbance at 570 nm. Quantification was performed using a standard curve obtained from known concentrations of (±)-6-Hydroxy-2,5,7,8-tetramethylchromane-2-carboxylic acid, also known as Trolox. Data were expressed as µM of Trolox equivalents.

### 4.10. Data and Statistical Analysis 

The boiling experiment was repeated five times on different days. Three of the experiments were used for TAC and TPC analysis. Data represent means and standard deviations from combined experiments. All kinetics of the release of selected compounds are shown as saturation curves. Areas under the curves (AUCs) were obtained from peaks observed from ^1^H NMR or LC-MS/MS spectra. The AUCs of the NMR peaks were measured manually. The times at which the maximum levels of metabolites were released (t_max_) were estimated by plotting the AUC at each time point. T_max_ values were calculated as 4× the time constant τ obtained from the one-phase association equation using GraphPad Prism software version 9.1.0.

## 5. Conclusions

In many communities around the world, natural health products, including herbal remedies, supplements, and traditional medicines, are commonly utilized for various health purposes. Although widely used, there is a lack of rigorous scientific analysis including characterization and standardization of preparations for their use in clinical trials. Here, we developed an NMR- and MS-based saturation kinetics model for the characterization and standardization of a decoction for a clinical trial. We followed the kinetics of the release of key groups of metabolites, the total phenolic content, and the total antioxidant capacity and found that 61 min of boiling of 24 g of BP in 250 mL of water generates the optimal decoction. Confirming the presence of Q-3O-ArRh and measuring TPC, TAC, and pH are recommended as qualitative and quantitative parameters to standardize the preparation of BP as a decoction for biological evaluation. This work represents the first metabolite kinetic-based study of a decoction formulation for in vivo use. As the interest in studying traditional medicines and decoctions increases, there is a need to use this type of approach to better understand the contents of the formulations and to standardize their preparation to optimize reproducibility.

## Figures and Tables

**Figure 1 ijms-25-05280-f001:**
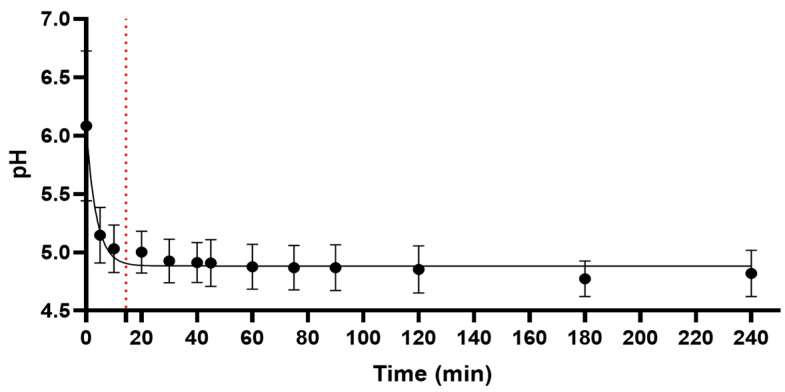
Kinetics of pH of *B. pinnatum* decoction during boiling. Data shown are means ± standard deviation (SD) from 5 independent experiments. After 14 min (shown by red dotted line), the pH of the decoction stabilized to an average of 4.8.

**Figure 2 ijms-25-05280-f002:**
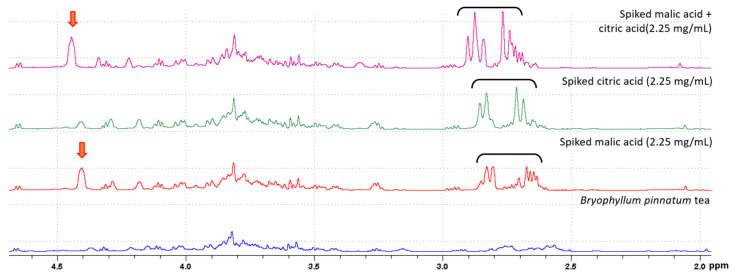
Identification of malic acid and citric acid in BP decoction as characterized by ^1^H NMR. The decoction was spiked with 2.25 mg/mL of either malic acid, citric acid, or both. An increase in peak intensity was observed in the chemical shift range of 2.38–2.85 ppm for both malic acid and citric acid as indicated by the black bracket. Additionally, malic acid had a unique singlet observed at 4.18 ppm denoted by the orange arrow.

**Figure 3 ijms-25-05280-f003:**
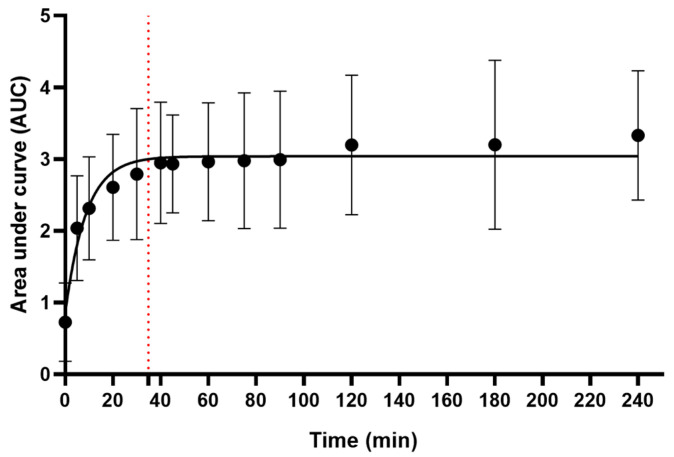
Combined kinetics of release of citric and malic acid into the *B. pinnatum* decoction shown as area under the curve versus boiling time. Within 35 min, citric acid and malic acid were maximally released into the decoction (t_max_ shown by red dotted line). Data shown are means ± SD from 5 independent experiments.

**Figure 4 ijms-25-05280-f004:**
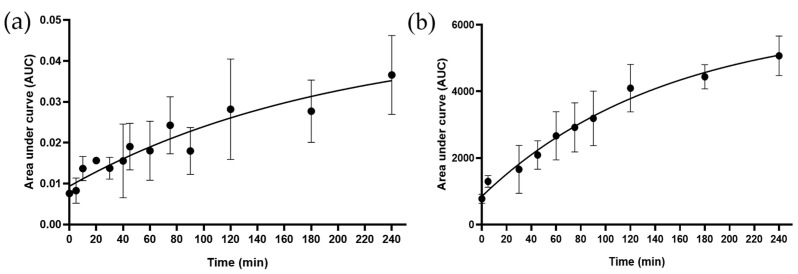
Kinetics of release of gallic acid into the *B. pinnatum* decoction as characterized by (**a**) ^1^H NMR and (**b**) LC-MS/MS. The saturation of gallic acid in solution was predicted to occur at approximately 13 and 10 h by ^1^H NMR and LC-MS/MS, respectively. ^1^H NMR and MS data are means ± SD from 3 and 5 independent experiments, respectively.

**Figure 5 ijms-25-05280-f005:**
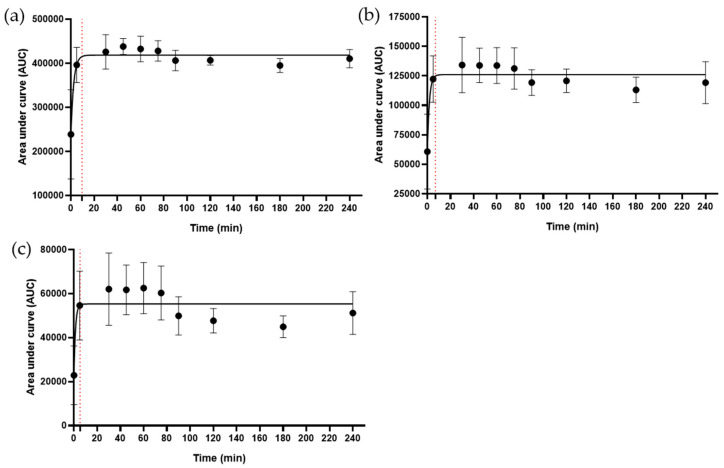
Kinetics of release of flavonoids conjugated with disaccharides into the *B. pinnatum* decoction were characterized by LC-MS/MS. (**a**) Q-3O-ArRh, (**b**) M-3O-ArRh, and (**c**) kappinatoside identified at *m*/*z* 579 [M−H]^−^, 595 [M−H]^−^, and 563 [M−H]^−^, respectively. Maximal release of these flavonoids was observed between 5 and 10 min (shown by red dotted lines). Data shown are means ± SD from 5 independent experiments.

**Figure 6 ijms-25-05280-f006:**
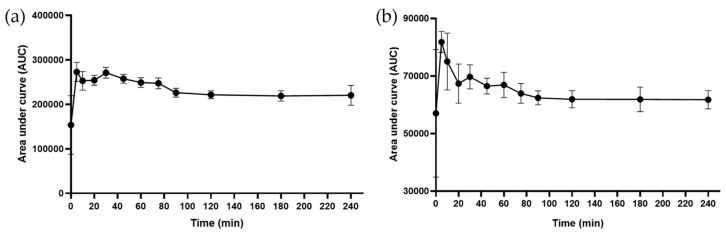
Kinetics of release of flavonoids conjugated with monosaccharides (**a**) myricitrin and (**b**) quercitrin into the *B. pinnatum* decoction as characterized by LC-MS/MS at *m*/*z* 463 [M−H]^−^ and 447 [M−H]^−^, respectively. Data shown are means ± SD from 5 independent experiments.

**Figure 7 ijms-25-05280-f007:**
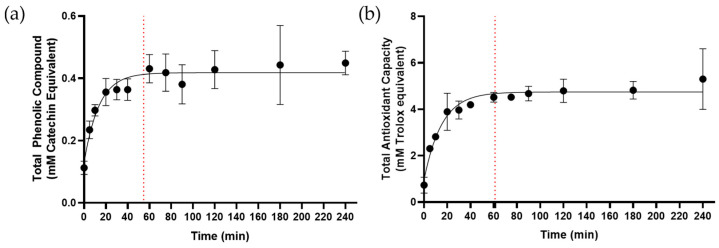
(**a**) Kinetics of release of total phenolic content into the *B. pinnatum* decoction. Maximal release was observed within 55 min at 0.4 mM catechin equivalent (shown by red dotted line). (**b**) Kinetics of release of total antioxidant capacity. Maximal release was observed within 61 min at 4.7 mM Trolox equivalent (shown by red dotted line). Data shown are means ± SD from 3 independent experiments.

**Figure 8 ijms-25-05280-f008:**
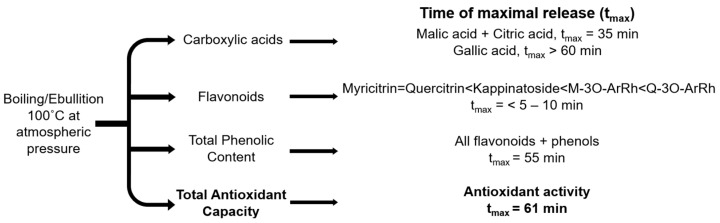
Key metabolic signature of *Bryophyllum pinnatum* decoction for clinical use. The kinetics of the total antioxidant capacity were used to define the duration of boiling of 61 min.

## Data Availability

The original contributions presented in the study are included in the article/Appendix A, and further inquiries can be directed to the corresponding author. The raw data supporting the conclusions of this article will be made available by the authors on request.

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
