# Peer review of "A Nuclear Magnetic Resonance (NMR)- and Mass Spectrometry (MS)-Based Saturation Kinetics Model of a Bryophyllum pinnatum Decoction as a Treatment for Kidney Stones"

_ijms, 2024, doi:10.3390/ijms25105280_

Round 1

Reviewer 1 Report

Comments and Suggestions for Authors

The article submitted for review concerns the analysis of the release of compounds from Bryophyllum pinnatum during boiling in water at 373K. 

The article was carefully prepared, the illustrations presented are clear and accurately illustrate the results and conclusions presented by the authors. 

My big concern is with the term 'tea' that runs through from the beginning. In relation to preparations made from plant substances, we can speak of infusions, decoctions, macerates, but the term tea is far from precise and, from the title, by the introduction may create interpretation problems for the reader. If the authors are talking about potential medical use, I suggest using pharmaceutical terminology.

Similarly, the first sentence of the abstract : "Bryophyllum pinnatum (BP) is a medicinal plant used to treat many conditions when taken as a leaf juice, leaves in capsules, as an ethanolic extract, and as a tea." does not sound like scientific language, but popular science. What are leaves in capsules?

Methods:

Were the time and place of harvesting the leaves the same?

Is the morphological evaluation of the leaves sufficient to be sure that the leaves are from the same variety?

How were the leaves fragmented? Cutting into small pieces is a vague concept and indicates low reproducibility. 

Was the volume of solvent taken from the system replenished after sampling?

 Lack of basic NMR analysis parameters.

Did the Authors not think to presaturate the water signal in the spectra? This would have had a beneficial effect on the readability of the spectra. 

Experiments, except the TAC and TPC, were repeated five times - how many times were TAC and TPC repeated?

Were only the measurements repeated or also the extraction process?

Results: 

Fig. 2 is illegible and needs improvement. A fragment of the spectrum analysed should be shown, not the entire range of chemical shift.

How was the area under the NMR signal measured? Was the signal integration manual or automatic?

Fig. 6 - caption. Flavonoid kinetics? 

Conclusions: 

The effect of raw material cooking length on the release of polyphenols, flavonoids is already widely reported in the literature. How does the raw material used in this study differ and were differences observed in relation to studies published to date? A discussion should be developed.

Author Response

April 28, 2024

Dear Editor,

We appreciate the comments from the reviewers and in the following text we will respond to each comment as requested.

Reviewer 1:

1.    “My big concern is with the term 'tea' that runs through from the beginning. In relation to preparations made from plant substances, we can speak of infusions, decoctions, macerates, but the term tea is far from precise and, from the title, by the introduction may create interpretation problems for the reader. If the authors are talking about potential medical use, I suggest using pharmaceutical terminology.”

We agree with this comment and have revised the text accordingly.  We changed the term “tea” to “decoction”, which is, as per the referee’s suggestion, a more pharmaceutical terminology to refer to the extracts resulting from prolonged boiling of the leaves over the 4-hour interval of the study.

2.    “Similarly, the first sentence of the abstract: "Bryophyllum pinnatum (BP) is a medicinal plant used to treat many conditions when taken as a leaf juice, leaves in capsules, as an ethanolic extract, and as a tea." does not sound like scientific language, but popular science. What are leaves in capsules?”

We agree with the reviewer’s assessment. Indeed, many studies that have used BP in the form of tablets are unclear with the contents and the formulation.  In a clinical trial by Simões-Wüst et al., (2018), tablets containing 50% BP were used and described as “170 mg of leave press juice, dried down to 17 mg by mixing with lactose; 100 mg dried B. pinnatum matter in 1 g”. Other trials describe it as 50% aqueous extract mixed into capsules (Betschart et al., 2013). Indeed, we needed to refer to these apparent “popular science” terms (e.g leaf juice, leaves in pills), already used in the literature, in order to lay the ground for the purpose of our work, which was to develop a more rigorous characterization based on clinically relevant molecular signatures of the decoction.

3.    “Were the time and place of harvesting the leaves the same?”

The leaves were grown under controlled light, humidity, and temperature, indoors at the Faculty of Agricultural and Environmental Sciences, McDonald Campus, McGill University. All the leaves used in the study were harvested from the same place. Two batches of leaves were obtained at different times, stored in a -20 freezer for one to three weeks before use. Each batch was thawed at room temperature for 30 minutes and then used for boiling experiments. We have added these statements to the methods for clarity. 

4.    “Is the morphological evaluation of the leaves sufficient to be sure that the leaves are from the same variety?”
Morphological evaluation of the leaves is the standard for their identification and collection. We included a photo of the leaves (Supplementary Figure 1) to demonstrate their morphology showing broad leaves with scalloped edges. Further evidence that the plant was indeed Bryophyllum pinnatum is found in the context of our study with the identification of its unique flavonoid, quercetin 3-O-α-L-arabinopyranosyl-(1 → 2)-α-L-rhamnopyranoside (Q-3O-ArRh), also referred to by Araújo et al. as BP-1 (Ref 36). 

5.    “How were the leaves fragmented? Cutting into small pieces is a vague concept and indicates low reproducibility.”

We agree and have now added details to the methods describing that the leaves were consistently fragmented into approximately 1 by 0.5 cm rectangular fragments in each experiment.

6.    “Was the volume of solvent taken from the system replenished after sampling?”

For sampling, 1 mL aliquots were taken at each of the 13 timepoints, representing a total of 13 mL from the original 250 mL of water used throughout the entire boiling kinetics experiments. This total volume represents a 5.2% loss of water during prolonged boiling, which is negligible and unlikely to affect the saturation kinetics. It is also important to note that the experiments were performed under reflux to avoid loss of water through evaporation.

7.    “Lack of basic NMR analysis parameters.”

We agree with the reviewer and have added more details to the methods section (section 4.6) including acquisition parameters such as pulse, locked solvent, relaxation delay, mixing time (tm) etc.  

8.    “Did the Authors not think to presaturate the water signal in the spectra? This would have had a beneficial effect on the readability of the spectra.”

We presaturated the water signal, but as the samples were not overly concentrated, we still saw the water signal but with little to no baseline distortion.

9.    “Experiments, except the TAC and TPC, were repeated five times - how many times were TAC and TPC repeated? Were only the measurements repeated or also the extraction process?”

The extraction process was repeated five times. However, given the limited availability of the content of the kits used in this study, TAC and TPC were measured in three sets of samples resulting from of three independent experiments. This has been clarified in the methods section (4.10). 

10.    “Fig. 2 is illegible and needs improvement. A fragment of the spectrum analysed should be shown, not the entire range of chemical shift.”

We agree and have updated the figure.

11.    “How was the area under the NMR signal measured? Was the signal integration manual or automatic?”

The area under the NMR signal was measured manually.

12. “Fig. 6 - caption. Flavonoid kinetics?”

 We agree and have revised the caption accordingly. 

13.    “The effect of raw material cooking length on the release of polyphenols, flavonoids is already widely reported in the literature. How does the raw material used in this study differ and were differences observed in relation to studies published to date? A discussion should be developed.”

We agree and have included some discussion on this subject. We observed that the flavonoid signature is stable over prolonged boiling at 100ËšC for up to 240 minutes. This is consistent with another study that showed higher temperatures to be more favorable for release of catechins and gallic acid.

Yours sincerely,

Indra R. Gupta
Professor
Dept of Pediatrics
Associate Member
Dept of Human Genetics and Experimental Medicine
McGill University

Reviewer 2 Report

Comments and Suggestions for Authors

The manuscript describes the investigation of the saturation kinetics of release of metabolites from tea made from boiling Bryophyllum pinnatum leaves using NMR and MS analyses, towards informing potential treatment of kidney stones.

Overall the manuscript is well prepared with adequate reproducibility and informative results.

It needs a separate Conclusion section, and clinical/practical implications.  

The title is misleading, because the work is not pharmacological study but measurement of release profile of some metabolites, with indication of clinical treatment of kidney stones. Please modify.

One aspect needs to be considered and discussed: water evaporation. 4 hour boiling is a long time and there is significant water loss if refluxing is not done properly. Have you quantified water loss? And water evaporation will cause concentration increase in a solution. Would this interfere with the result interpretation and the current conclusion? It does not look like any refluxing has been done though.

Common knowledge is that boiling tea at high temperature for extended time destroys healthful compounds. Can you comment on this and compare the release profile with the current extraction methods?

Was pH measured for -20 C aliquots or for after it was thawed to RT?

Fig. 7A, data point 240 min is missing.

Line 259 suggest adding a citation that explores structure-property relationship of phenolic compounds of tea Systematic characterisation of the structure and radical scavenging potency of Pu'Er tea () polyphenol theaflavin 2019.

Author Response

April 28, 2024

Dear Editor,

We appreciate the comments from the reviewers and in the following text we will respond to each comment as requested.

Reviewer 2:
1.    “It needs a separate Conclusion section, and clinical/practical implications.”

We agree and have added a conclusion section and indicated that this method can be used to standardize other preparations and traditional medicines. 

2.    “The title is misleading, because the work is not pharmacological study but measurement of release profile of some metabolites, with indication of clinical treatment of kidney stones. Please modify.”

We agree and have revised the title according to this suggestion to: “A Nuclear Magnetic Resonance (NMR) and Mass Spectrometry (MS)-based saturation kinetics model defines a Bryophyllum pinnatum decoction as a treatment for kidney stones”. 

3.    “One aspect needs to be considered and discussed: water evaporation. 4-hour boiling is a long time and there is significant water loss if refluxing is not done properly. Have you quantified water loss? And water evaporation will cause concentration increase in a solution. Would this interfere with the result interpretation and the current conclusion? It does not look like any refluxing has been done though.”

We thank the reviewer for this comment, which was partially addressed in our response to Reviewer 1. As indicated in the methods section (4.2), the decoction was performed under reflux using a condenser kept at 10 oC using a chiller. Therefore, there was no water volume loss through evaporation during 4 hours of boiling. 

4.    “Common knowledge is that boiling tea at high temperature for extended time destroys healthful compounds. Can you comment on this and compare the release profile with the current extraction methods?”

The metabolic signature that we defined includes “healthful compounds” that are are effective to treat kidney stones. The saturation kinetic profiles indicate that they remain stable following their release during the 4-hour boiling interval. We have added a reference that addressed the stability of free quercetin at 100 oC leading to degradation metabolites. Our findings are consistent with others that showed  higher temperatures to be favorable for phenolic compounds and flavonoid release. The boiling process is also advantageous because it will inactivate any contamination with bacteria, viruses, and/or protozoa.

5.    “Was pH measured for -20 C aliquots or for after it was thawed to RT?”

All measurements, including pH, were done after the aliquots were thawed to RT (section 4.1). 

6.    “Fig. 7A, data point 240 min is missing.”

We thank the reviewer for raising this point. The graph has now been corrected. All graphs in the paper include data acquired over the full 4 h period of the kinetic analysis. 

7.    “Line 259 suggest adding a citation that explores structure-property relationship of phenolic compounds of tea Systematic characterisation of the structure and radical scavenging potency of Pu'Er tea () polyphenol theaflavin 2019.”

We agree and we thank the reviewer for recommending this important citation on the antioxidant properties of metabolites in Pu’Er tea (Organic & Biomolecular Chemistry 2019, 17, 9942-9950). It is now included in our Discussion section. 

Yours sincerely,

Indra R. Gupta
Professor
Dept of Pediatrics
Associate Member
Dept of Human Genetics and Experimental Medicine
McGill University

Round 2

Reviewer 2 Report

Comments and Suggestions for Authors

All the issues have been addressed.

Control to make sure all the references are in the same style, especially the author names of the newly added ones. 

Author Response

We thank the reviewer for picking up the error in the newly added references. We have revised this in the current version.

Thank you

Indra Gupta